# Isolation and characterization of *Septuagintavirus*; a novel clade of *Escherichia coli* phages within the subfamily *Vequintavirinae*

Adrián Cortés-Martín,[1] Colin Buttimer,[1] Nadiia Pozhydaieva,[2] Frank Hille,[3] Hiba Shareefdeen,[1] Andrei S. Bolocan,[1] Lorraine A. Draper,[1] Andrey N. Shkoporov,[1] Charles M. A. P. Franz,[3] Katharina Höfer,[2,4] R. Paul Ross,[1] Colin Hill[1,5]

**ABSTRACT** *Escherichia coli* is a commensal inhabitant of the mammalian gut microbiota, frequently associated with various gastrointestinal diseases. There is increasing interest in comprehending the variety of bacteriophages (phages) that target this bacterium, as such insights could pave the way for their potential use in therapeutic applications. Here, we report the isolation and characterization of four newly identified *E. coli* infecting tailed phages (W70, A7-1, A5-4, and A73) that were found to constitute a novel genus, *Septuagintavirus*, within the subfamily *Vequintavirinae*. Genomes of these phages ranged from 137 kbp to 145 kbp, with a GC content of 41 mol%. They possess a maximum nucleotide similarity of 30% with phages of the closest phylogenetic genus, *Certrevirus*, while displaying limited homology to other genera of the *Vequintavirinae* family. Host range analysis showed that these phages have limited activity against a panel of *E. coli* strains, infecting 6 out of 16 tested isolates, regardless of their phylotype. Electrospray ionization-tandem mass spectrometry (ESI-MS/MS) was performed on the virion of phage W70, allowing the identification of 28 structural proteins, 19 of which were shared with phages of other genera of *Vequintavirinae* family. The greatest diversity was identified with proteins forming tail fiber structures, likely indicating the adaptation of virions of each phage genus of this subfamily for the recognition of their target receptor on host cells. The findings of this study provide greater insights into the phages of the subfamily *Vequintavirinae*, contributing to the pool of knowledge currently known about these phages.

**IMPORTANCE** *Escherichia coli* is a well-known bacterium that inhabits diverse ecological niches, including the mammalian gut microbiota. Certain strains are associated with gastrointestinal diseases, and there is a growing interest in using bacteriophages, viruses that infect bacteria, to combat bacterial infections. Here, we describe the isolation and characterization of four novel *E. coli* bacteriophages that constitute a new genus, *Septuagintavirus*, within the subfamily *Vequintavirinae*. We conducted mass spectrometry on virions of a representative phage of this novel clade and compared it to other phages within the subfamily. Our analysis shows that virion structure is highly conserved among all phages, except for proteins related to tail fiber structures implicated in the host range. These findings provide greater insights into the phages of the subfamily *Vequintavirinae*, contributing to the existing pool of knowledge about these phages.

**KEYWORDS** isolation, characterization, bacteriophage, *Escherichia coli*, *Vequintavirinae*, virion, mass-spectrometry

*E*scherichia coli is a well-known Gram-negative bacterium that thrives in diverse ecological niches and holds considerable significance in animal and human health. As a commensal member of the gut microbiota in mammals, *E. coli* stands out as a prominent facultative anaerobic bacterium within the intestinal environment. The

Address correspondence to Colin Hill, c.hill@ucc.ie.

Adrián Cortés-Martín and Colin Buttimer contributed equally to this article. The author order was determined by a coin toss.

The authors declare no conflict of interest.

See the funding table on p. 14.

species exhibits an extensive genetic substructure, categorized into distinct phylogroups (A, B1, B2, C, D, E, F, and clade I), each with unique phenotypic and genotypic characteristics (1). While most *E. coli* strains are benign, others can act as opportunistic pathogens, contributing to gastrointestinal diseases such as diarrhea or dysentery (2). They have been linked to the pathogenesis of inflammatory bowel diseases (IBD) (3), and they are also implicated in extraintestinal illnesses, including pneumonia, urinary tract infections, peritonitis, and bacteremia (4, 5). The conventional treatment for *E. coli* generally involves antibiotics. However, the excessive and inappropriate use of these drugs has led to the emergence of antibiotic resistance among bacteria that impact human health. This, in turn, substantially burdens healthcare systems, leading to increased mortality rates and costs (6, 7).

Bacteriophages (phages) are viruses that specifically infect bacteria. The estimated ratio of phages to bacteria in the human gut is thought to vary between 1:1 and 1:100 (8). There is growing interest in their development as tools for engineering microbial gut communities (9), and as an alternative therapy to combat bacterial diseases. Several *E. coli* phages have been reported for use in phage therapy targeting various pathogenic *E. coli* strains and as adjuncts to probiotics with promising effects (10–16). Nevertheless, achieving desired outcomes requires a deeper comprehension of the interactions between phages and their hosts and the continuous isolation of new phages. The diversity of bacterial species presents a significant challenge for the successful application of phages, emphasizing the ongoing necessity for isolating lytic phages. Assigning phages to distinct taxonomic groups is a fundamental step following their discovery.

The International Committee on Taxonomy of Viruses (ICTV) currently classifies *Escherichia* phages of the class *Caudoviricetes* (tailed phages) into a minimum of 37 genera (17). At present, the subfamily *Vequintavirinae* contains six genera: *Vequintavirus, Avunavirus, Certrevirus, Seunavirus, Mydovirus,* and *Henunavirus*, containing a total of 26 species. To date, phages of this subfamily infect bacterial species of the Order Enterobacterales, often found associated with mammalian intestines, such as *Escherichia*, *Cronobacter, Salmonella, Klebsiella, Raoultella,* and *Proteus* (18–22). Additionally, they target species linked to plant environments, such as *Erwinia* and *Pectobacterium* (23, 24). Phages of this subfamily possess a myovirus morphology with genomes ranging from 120 to 151 kbp (21, 24). These phages employ various host cell receptors to interact with their host. Specifically, *Vequintavirus* phages utilize the enterobacterial common antigen—a carbohydrate antigen present in the outer membrane of many Enterobacterales species—as their primary receptor for host infection. This feature is believed to explain why phages of the *Vequintavirus* possess a broad host range (25). In contrast, other *Vequintavirinae* genera, such as *Certrevirus*, have been shown to use the flagella of their host as their primary receptor, with these phages typically possessing a much narrower host range (22, 26).

Here, we present the isolation, characterization, and taxonomic classification of four novel lytic *E. coli* phages (W70, A7-1, A5-4, and A73) that infect *E. coli* K12 MG1655. Phylogenetic analysis of their genomes has led to the establishment of a new genus, *Septuagintavirus*, comprising two new species within the subfamily *Vequintavirinae*. Progress in the characterization and classification of novel phages is important to continue our efforts to improve phage therapy as a possible alternative to fight antibiotic resistance.

## RESULTS AND DISCUSSION

### Isolation of *Escherichia* phages and their characterization

*E. coli* phages were isolated from wastewater (W70) and animal feces (A7-1, A5-4, and A73) collected on a multi-species farm in Munster (Ireland) using *E. coli* MG-1655 as a host. All phages formed clear plaques with an approximate diameter of 1–3 mm (overlay 0.4% agar wt/vol) when plated on the lawn of the host strain (Fig. S1A through E). The host range analysis indicated some activity against other *E. coli* strains isolated from the feces of individuals with IBD (Table S1 and S3; Fig. S1F). Sixteen different *E. coli*

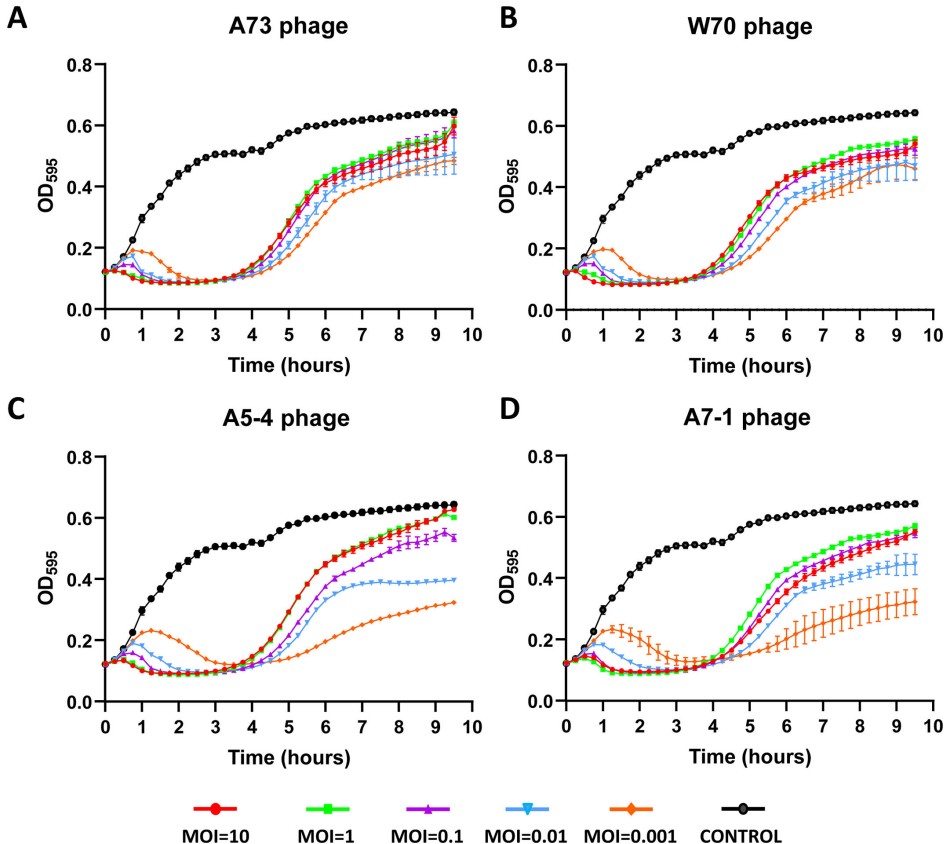

**FIG 1** Bacteriophage kill-curves of *E. coli* K12 MG-1655 in response to infection by novel isolated *E. coli* phages. The host was individually infected with the four phages: (A) A73, (B) W70, (C) A5-4, and (D) A7-1 at different MOIs: 10 (red), 1 (green), 0.1 (purple), 0.01 (blue), and 0.001 (orange). Growth control (only bacteria) is represented with black color. All samples were analyzed in triplicate. Results are expressed as mean values ± SD.

strains representing various phylotypes (A, B1, B2, D, and E) were examined, revealing no discernible correlation between *E. coli* phylotypes and phage lytic activity. The A5-4 and A7-1 phages formed plaques when infecting the *E. coli* 311IE2-3 strain (phylotype B2). However, they did not yield plaques in other strains of the same phylotype. The A73 and W70 phages produced only tiny halos at high titer within this strain, failing to produce proper plaques. For the W70 phage, plaques were observed when infecting the *E. coli* 523IE5-40 strain (phylotype D); however, it did not produce plaques in other strains with the same phylotype. The rest of the phages produced only zones of clearing at high titer in this strain. Apart from these two strains, similar tiny halos were detected without plaque formation when high phage titers were tested in four other strains belonging to different phylotypes: A, B1, B2, and D (Table S3). In summary, 10 out of the 16 tested strains displayed no sensitivity to any of the phages, and notably, none of the tested strains belonging to phylotype E showed any susceptibility to the phages. Further inspection of the host range with additional genera from the family *Enterobacteriaceae* (*Klebsiella pneumoniae*, *Enterobacter aerogenes*, *Salmonella enterica* serovar typhimurium, *Proteus vulgaris,* and *Citrobacter braakii*) showed no sensitivity to infection by these phages (Table S3). We also performed an *in silico* host range prediction using the iPHoP tool and the genomes of these phages. The random forest classifier within this tool identified their host range primarily within the genus *Escherichia* (Table S4).

The lysis profile and the potential development of phage resistance were evaluated by quantifying host bacterial culture density at $OD_{595}$ following phage infection at various MOIs (Fig. 1). The lysis of host cultures occurred rapidly and was contingent upon the MOI for all phages. In all cases, higher MOIs (ranging from 0.1 to 10) led to

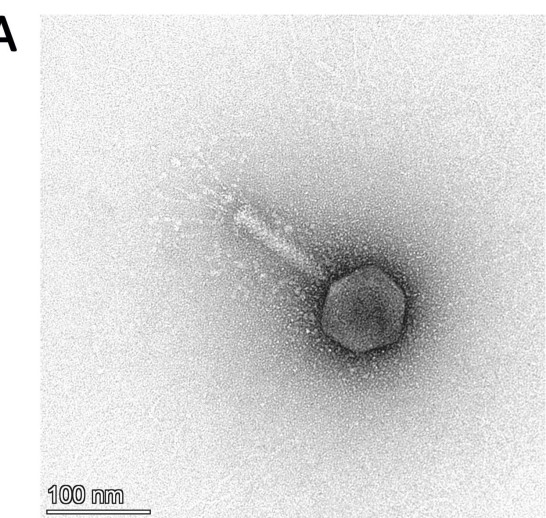

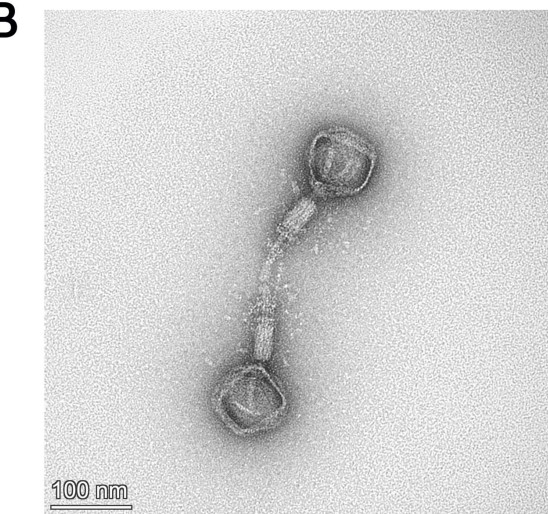

**FIG 2** Transmission electron micrographs of *Escherichia* phage W70 stained with 2% (wt/vol) uranyl acetate. (A) Uncontracted phage W70 virion particle and (B) two particles with empty capsids and contracted tail sheath. The scale bar represents 100 nm.

a decrease in $OD_{595}$ in less than an hour due to the lytic activity of the phages. Slight bacterial growth was observed with the lowest MOIs (0.001–0.01) during the initial hour. However, due to phage replication and their associated lytic activity, $OD_{595}$ eventually decreased, returning to the initial absorbance values. After 3–4 hours of infection, the $OD_{595}$ began to increase, indicating the appearance of resistance to all the phages tested. Although the behavior of the host exhibited a similar response to all four phages, certain differences were noted. For instance, when the host was infected with A73 and W70 phages, bacterial growth after becoming phage-resistant was very similar, independent of the initial MOI (Fig. 1A and B). However, when the host was infected with A5-4 and A7-1 phages, lower MOIs appeared to result in less efficient recovery than higher MOIs (Fig. 1C and D).

We were able to purify phage lysates of phage W70 by CsCl gradient (Fig. S2A). TEM analysis of such a preparation showed that phage W70 possesses a myovirus A1 morphotype (Fig. 2) (27). The phage consists of an icosahedral head (68.65 ± 3 nm in diameter, $n = 12$) with clearly distinguishable hexagonal outlines and a contractile tail (120.36 ± 4.12 nm × 25.39 nm, $n = 12$) that includes a bundle of thin and flexible tail fibers.

**TABLE 1** Properties of the four phages belonging to the new genus *Septuagintavirus*

| Phage | INSDC accession number | Genome size (bp) | Number of ORFs | Number of tRNA genes | GC content (mol%) | Nucleotide homology (%)[a] | Homologous proteins (%)[a] |
|---|---|---|---|---|---|---|---|
| W70 | OP778610 | 137,323 | 251 | 5 | 41 | 100 | 100 |
| A73 | OP778609 | 142,009 | 264 | 5 | 41 | 96 | 96 |
| A7-1 | OP795442 | 144,828 | 273 | 5 | 41 | 91 | 96 |
| A5-4 | OP744025 | 145,365 | 276 | 5 | 41 | 92 | 95 |

[a]*E. coli* phage W70 was used as a reference for nucleotide homology and homologous protein data. INSDC: International Nucleotide Sequence Database Collaboration; ORF: Open reading frame.

## Genomes of the novel isolated *E. coli* phages

The genomes obtained for the *Escherichia* phages ranged between 137,323 and 145,365 bp (200x–418x coverage) with a GC content of 41 mol% (Table 1). Analyzing the nucleotide similarity between genomes revealed homology across all phage isolates at the nucleotide level, with a shared nucleotide similarity between genomes exceeding 90% (Fig. 3). Following the International Committee on Taxonomy of Viruses (ICTV) guidelines, a shared nucleotide similarity higher than 95% between phages allows them to be assigned to the same species (28). As a result, *Escherichia* phages A73 and A5-4 could be classed as strains of phages W70 and A7-1, respectively (Fig. 3).

The number of open reading frames (ORFs) found on the genomes of these phages ranged from 251 to 276, with each phage also possessing five tRNA genes (Table 1). To explore the common protein content among the four phages, we conducted a pangenome analysis of their proteins, employing an identity and a coverage threshold of 30% and 70%, respectively. These proteins were placed into 311 orthologous groups (OGs), and the shared proteins, forming the core proteome across these genomes, constituted 223 OGs, representing 83% of the total. Each phage exhibited 8 to 10 unique proteins not shared with other isolates.

Functional assignments were successfully attributed to only 26% of the OGs, and there was notable overlap among various OGs annotated with similar functions. Our annotation endeavors grouped OGs into five primary functional categories: host lysis, DNA-related processes, transcriptional regulation, virion assembly, and miscellaneous functions (Fig. 4; Table S5). No OGs were identified to encode integrase, excisionase, or repressor proteins, indicating that these phages likely follow an exclusively lytic lifestyle.

| Phage | Genome | 1 | 2 | 3 | 4 | 5 | 6 | 7 | 8 | 9 | 10 | 11 | 12 | 13 | 14 | 15 | 16 | 17 | 18 | 19 | 20 | 21 | 22 | 23 | Genera |
|---|---|---|---|---|---|---|---|---|---|---|---|---|---|---|---|---|---|---|---|---|---|---|---|---|---|
| *Escherichia* phage A5-4 | 1 | 100 | 95 | 93 | 90 | 29 | 28 | 29 | 28 | 29 | 6 | 6 | 6 | 6 | 7 | 8 | 8 | 7 | 7 | 7 | 10 | 10 | 9 | 9 | *Septuagintavirus* |
| *Escherichia* phage A7-1 | 2 | 95 | 100 | 92 | 90 | 28 | 28 | 28 | 28 | 28 | 5 | 6 | 6 | 6 | 6 | 7 | 7 | 7 | 7 | 7 | 10 | 10 | 8 | 9 | |
| *Escherichia* phage A73 | 3 | 93 | 92 | 100 | 95 | 29 | 29 | 29 | 29 | 29 | 6 | 6 | 6 | 6 | 7 | 7 | 8 | 7 | 8 | 7 | 10 | 10 | 9 | 9 | |
| *Escherichia* phage W70 | 4 | 90 | 90 | 95 | 100 | 30 | 29 | 29 | 29 | 29 | 5 | 6 | 6 | 6 | 7 | 7 | 8 | 7 | 7 | 7 | 10 | 10 | 9 | 9 | |
| *Pectobacterium* phage phiTE | 5 | 29 | 28 | 29 | 30 | 100 | 55 | 60 | 60 | 60 | 4 | 5 | 4 | 5 | 6 | 9 | 8 | 8 | 8 | 8 | 11 | 11 | 9 | 9 | *Certrevirus* |
| *Cronobacter* phage CR9 | 6 | 28 | 28 | 29 | 29 | 55 | 100 | 68 | 69 | 69 | 4 | 4 | 4 | 4 | 6 | 11 | 10 | 10 | 10 | 10 | 11 | 11 | 10 | 11 | |
| *Cronobacter* phage PBES_02 | 7 | 28 | 28 | 29 | 29 | 60 | 68 | 100 | 90 | 91 | 4 | 4 | 4 | 4 | 7 | 10 | 10 | 10 | 10 | 11 | 11 | 11 | 11 | 10 | |
| *Cronobacter* phage CR3 | 8 | 28 | 28 | 29 | 29 | 60 | 69 | 90 | 100 | 95 | 4 | 4 | 4 | 4 | 6 | 10 | 11 | 10 | 10 | 10 | 11 | 10 | 11 | 11 | |
| *Cronobacter* phage CR8 | 9 | 29 | 28 | 29 | 29 | 60 | 69 | 91 | 95 | 100 | 4 | 4 | 4 | 4 | 6 | 11 | 11 | 10 | 10 | 10 | 11 | 10 | 11 | 11 | |
| *Escherichia* phage FV3 | 10 | 6 | 5 | 6 | 5 | 4 | 4 | 4 | 4 | 4 | 100 | 89 | 89 | 89 | 10 | 8 | 7 | 12 | 12 | 12 | 14 | 14 | 12 | 13 | *Vequintavirus* |
| *Escherichia* phage V5 | 11 | 6 | 6 | 6 | 6 | 5 | 4 | 4 | 4 | 4 | 89 | 100 | 93 | 92 | 11 | 8 | 7 | 12 | 12 | 12 | 13 | 13 | 12 | 12 | |
| *Escherichia* phage JES2013 | 12 | 6 | 6 | 6 | 6 | 4 | 4 | 4 | 4 | 4 | 89 | 93 | 100 | 93 | 10 | 7 | 7 | 12 | 12 | 12 | 13 | 13 | 12 | 13 | |
| *Escherichia* phage FFH2 | 13 | 6 | 6 | 6 | 6 | 5 | 4 | 4 | 4 | 4 | 89 | 92 | 93 | 100 | 11 | 7 | 7 | 12 | 12 | 13 | 13 | 13 | 13 | 12 | |
| *Escherichia* phage Av-05 | 14 | 7 | 6 | 7 | 7 | 6 | 6 | 7 | 6 | 6 | 10 | 11 | 10 | 11 | 100 | 16 | 16 | 25 | 25 | 24 | 30 | 30 | 27 | 28 | *Avunavirus* |
| *Erwinia* phage pEp_SNUABM_01 | 15 | 8 | 7 | 7 | 7 | 9 | 11 | 10 | 10 | 11 | 8 | 8 | 7 | 7 | 16 | 100 | 70 | 29 | 28 | 28 | 28 | 28 | 29 | 30 | *Henunavirus* |
| *Erwinia* phage Hena1 | 16 | 8 | 7 | 8 | 8 | 8 | 10 | 10 | 11 | 11 | 7 | 7 | 7 | 7 | 16 | 70 | 100 | 28 | 28 | 27 | 27 | 27 | 29 | 29 | |
| *Raoultella* phage Ro1 | 17 | 7 | 7 | 7 | 7 | 8 | 10 | 10 | 10 | 10 | 12 | 12 | 12 | 12 | 25 | 29 | 28 | 100 | 76 | 77 | 46 | 45 | 46 | 47 | *Mydovirus* |
| *Klebsiella* phage vB_KpnM_KB57 | 18 | 7 | 7 | 8 | 7 | 8 | 10 | 10 | 10 | 10 | 12 | 12 | 12 | 12 | 25 | 28 | 28 | 76 | 100 | 85 | 46 | 46 | 49 | 51 | |
| *Klebsiella* phage vB_KpnM_BIS47 | 19 | 7 | 7 | 7 | 7 | 8 | 10 | 10 | 10 | 10 | 12 | 12 | 12 | 12 | 24 | 28 | 27 | 77 | 85 | 100 | 46 | 45 | 48 | 50 | |
| *Salmonella* phage PVPSE1 | 20 | 10 | 10 | 10 | 10 | 11 | 11 | 11 | 10 | 11 | 14 | 13 | 13 | 13 | 30 | 28 | 27 | 46 | 46 | 46 | 100 | 94 | 64 | 64 | *Seunavirus* |
| *Salmonella* phage SSE121 | 21 | 10 | 10 | 10 | 10 | 11 | 11 | 11 | 10 | 10 | 14 | 13 | 13 | 13 | 30 | 28 | 27 | 45 | 46 | 45 | 94 | 100 | 63 | 63 | |
| *Cronobacter* phage vB_CsaM_GAP31 | 22 | 9 | 8 | 9 | 9 | 9 | 10 | 11 | 11 | 11 | 12 | 12 | 12 | 13 | 27 | 29 | 29 | 46 | 49 | 48 | 64 | 63 | 100 | 82 | |
| *Escherichia* phage 4MG | 23 | 9 | 9 | 9 | 9 | 9 | 11 | 10 | 11 | 11 | 13 | 12 | 13 | 12 | 28 | 30 | 29 | 47 | 51 | 50 | 64 | 63 | 82 | 100 | |

**FIG 3** Heatmap showing the percentage of nucleotide similarity of *Escherichia* phages isolated in this study and other closer phylogenetic members of the subfamily *Vequintavirinae* as calculated with VIRIDIC (29).

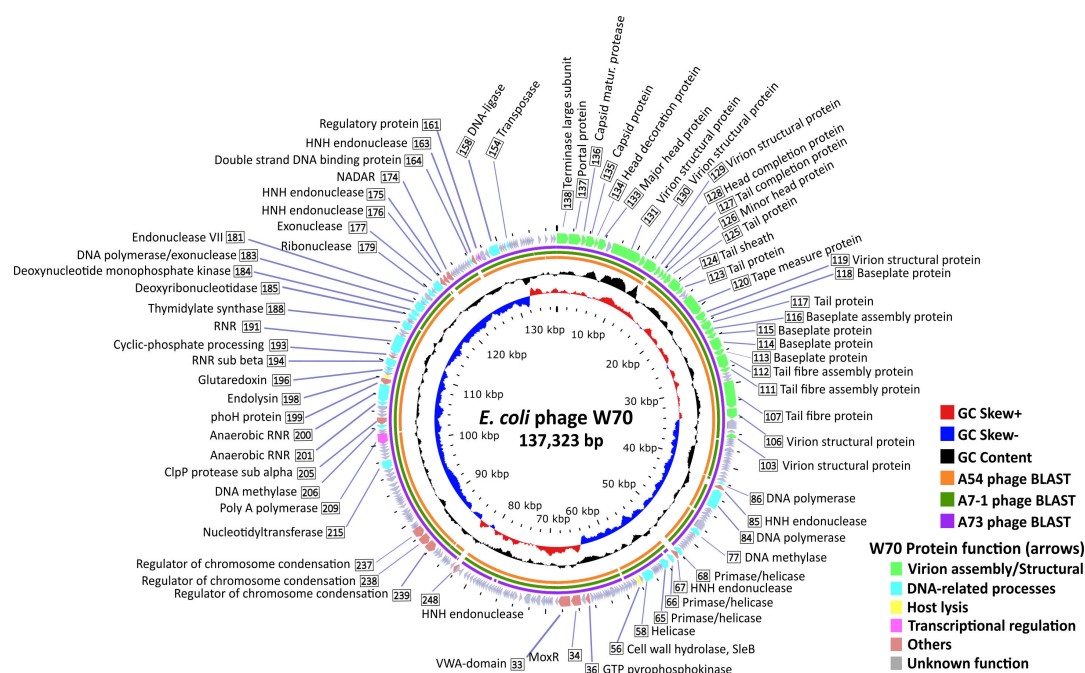

**FIG 4** Circular genome map of the *E. coli* phage W70 serving as a representative of the novel isolated *E. coli* phages. The innermost ring represents the GC skew (red for the positive strand and blue for the negative strand) while the central ring (black) shows the GC content. BLAST alignment was performed for sequencing similarity comparison versus A54 phage (orange ring), A7-1 phage (dark green ring), and A73 phage (purple ring). The outermost circle displays the coding genes (CDS) for W70 phage with predicted functions as labels. The coloration of the CDS corresponds to their general functions as indicated in the legend. Genes whose function could not be determined are colored gray and remain unlabeled.

## Phylogenetic position of the newly identified *Septuagintavirus* genus

These *Escherichia* phages did not fall into any phage genus currently recognized by the ICTV. A phylogenetic analysis VICTOR revealed that these isolated phages formed a distinct clade within the subfamily *Vequintavirinae*, positioned closely to *Certrevirus* phages (Fig. 5). Analysis of these genomes using BLASTn with the nt database showed that phages of the genus *Certrevirus* were the closest relatives but only shared a maximum nucleotide similarity of 30% (Fig. 3). Notably, *Escherichia* phages W70, A5-4, A7-1, and A73 shared a genome nucleotide similarity of ≥90%. These findings indicate

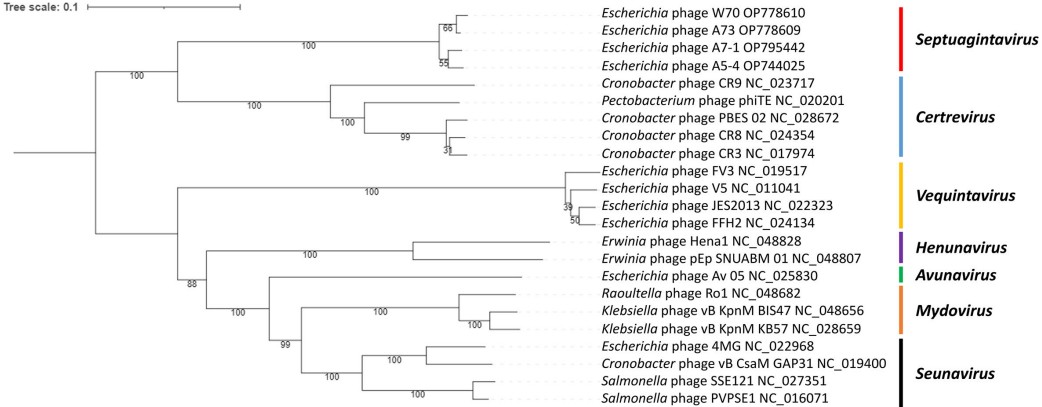

**FIG 5** Amino acid VICTOR-generated phylogenomic Genome-BLAST Distance Phylogeny (GBDP) tree inferred using the formula D4 and yielding an average support of 86% (30). The phylogram includes the *Escherichia* phages isolated in this study and other members of the subfamily *Vequintavirinae*. The genus (if allocated) of phages in the analysis is illustrated. Branch support was inferred from 100 pseudo-bootstrap replicates.

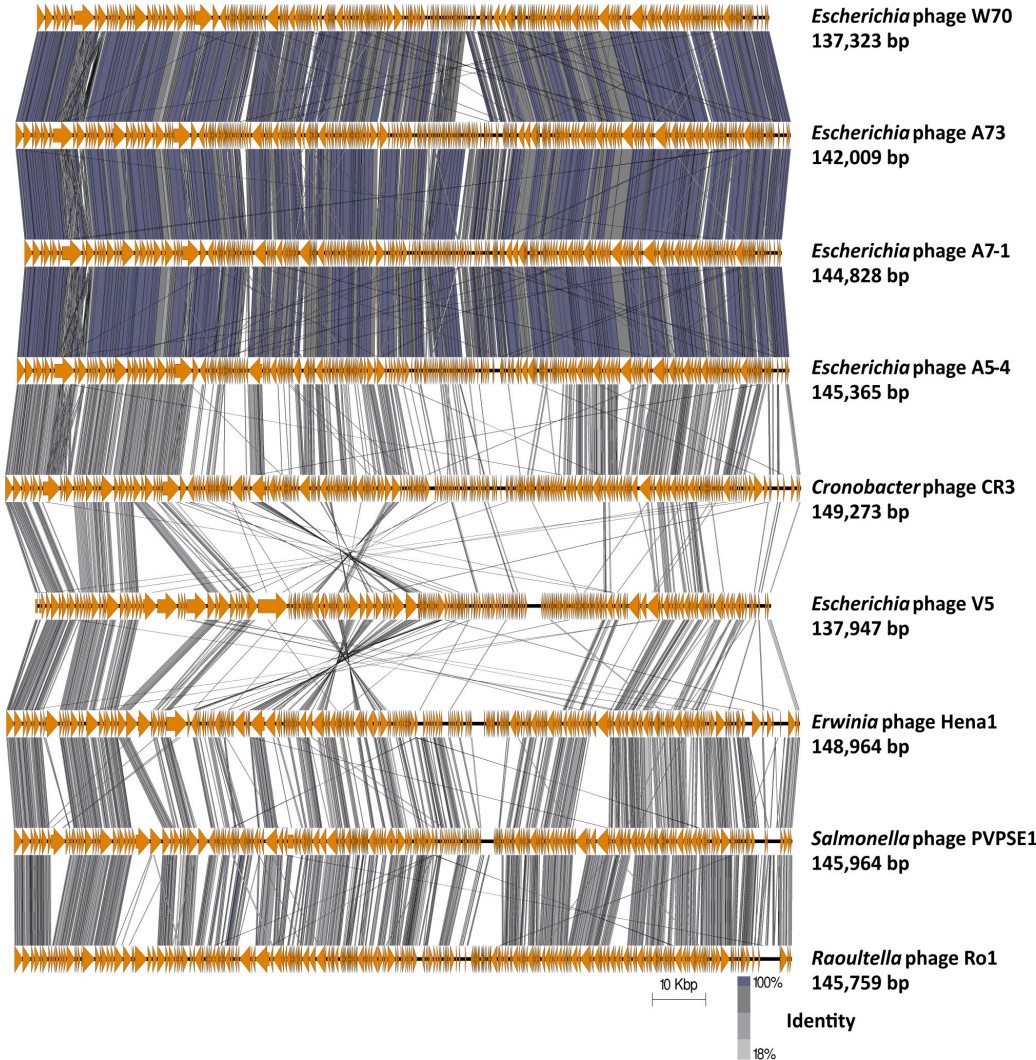

**FIG 6** Genome map comparison of the genomes of *Escherichia* phages isolated in this study and other members of the subfamily *Vequintavirinae* employing tBLASTx with easyfig (31). The genome maps display arrows indicating the locations and orientation of ORFs among different phage genomes. The large terminase was set as the first gene among all genomes.

that these phages meet the criteria for a novel phage genus (28), which we termed *Septuagintavirus* being derived from the name given to the first isolated example of these phages, namely *Escherichia* phage W70. Gene synteny was highly conserved between members of the new *Septuagintavirus* genus, and significant gene synteny was observed between *Septuagintavirus* and *Certrevirus* clades, as seen with *Cronobacter* phage CR3 and the phages of this study (Fig. 6).

## Structural proteome analysis of phage W70 particles

The structural proteins of the W70 phage were analyzed using electrospray-tandem mass spectrometry (ESI-MS/MS), extracted from a concentrated and purified sucrose gradient (Fig. S2B). This analysis successfully identified 28 virion proteins for which a number could be assigned a putative role in virion morphogenesis. Seven contributed to capsid formation, six were involved in tail formation, five were associated with baseplate formation, and two were likely engaged in tail fiber formation (Table 2). However, *in silico* analysis of the remaining eight structural proteins did not reveal a putative role. Among these eight, three proteins (W70_237, W70_238, and W70_239) were annotated with the potential function of a chromosome condensation protein (IPR009091). Similar proteins

**TABLE 2** Virion proteins identified in novel *Escherichia* phage W70 using electrospray-tandem mass spectrometry (ESI-MS/MS)[a]

| ORF | Predicted function | Phrog | Protein molecular weight (kDa) | Total spectrum count |
|---|---|---|---|---|
| W70_106 | Virion structural protein | phrog_15692 | 44 | 115 |
| W70_107 | Tail fiber protein | phrog_4549 | 112 | 996 |
| W70_112 | Tail fiber protein | phrog_12846 | 59 | 659 |
| W70_113 | Baseplate protein | phrog_4056 | 24 | 173 |
| W70_114 | Baseplate protein | phrog_6 | 54 | 212 |
| W70_115 | Baseplate protein | phrog_4233 | 20 | 67 |
| W70_116 | Putative baseplate assembly protein | - | 25 | 82 |
| W70_117 | Tail protein | phrog_196 | 35 | 68 |
| W70_118 | Baseplate protein | phrog_349 | 14 | 64 |
| W70_119 | Virion structural protein | phrog_5197 | 32 | 187 |
| W70_120 | Putative tape measure protein | phrog_5255 | 88 | 274 |
| W70_123 | Tail protein | phrog_280 | 17 | 220 |
| W70_124 | Tail sheath | phrog_205 | 51 | 311 |
| W70_125 | Tail protein | phrog_4528 | 25 | 61 |
| W70_126 | Minor head protein | phrog_4185 | 17 | 62 |
| W70_127 | Tail completion or Neck1 protein | phrog_222 | 17 | 28 |
| W70_128 | Head completion protein | phrog_2663 | 21 | 104 |
| W70_129 | Virion structural protein | phrog_5073 | 53 | 728 |
| W70_130 | Virion structural protein | phrog_4998 | 24 | 114 |
| W70_131 | Virion structural protein | phrog_3904 | 129 | 2032 |
| W70_133 | Major head protein | phrog_29 | 37 | 1596 |
| W70_134 | Head decoration-protein | phrog_2680 | 15 | 819 |
| W70_135 | Capsid protein | phrog_2970 | 40 | 913 |
| W70_136 | Capsid maturation protease | phrog_1530 | 21 | 40 |
| W70_137 | Portal protein | - | 56 | 467 |
| W70_237 | Regulator of chromosome condensation 1 | phrog_4219 | 43 | 233 |
| W70_238 | Regulator of chromosome condensation 1 | phrog_4219 | 40 | 196 |
| W70_239 | Regulator of chromosome condensation 1 | phrog_4219 | 41 | 141 |

[a]ORF: Open reading frame; PHROG: Protein orthologous groups.

found in other phage virions are suspected to play a role in encapsulating genomic DNA within the phage capsid (24).

Virion structural proteins of phage W70 were inspected through SDS-PAGE, and several of the identified proteins by ESI-MS/MS were discernible. This was evident as the molecular weights of the identified proteins matched the molecular weight of the observed protein bands (Fig. S3). These proteins included the major constituents of the phage W70 virion, including the major head protein (W70_133), the portal protein (W70_137), the capsid and capsid decoration proteins (W70_134, W70_135), tail fiber and sheath proteins (W70_107, W70_112, and W70_124), tape measure protein (W70_120), baseplate protein (W70_114) and virion structural proteins of unknown function (W70_106, W70_129, and W70_131). The presence of these proteins following SDS-PAGE suggests their high abundance in the formation of the phage W70 virion, and also aligns with the highest peptide spectrum counts obtained from mass spectrometry analysis.

Most of these structural proteins are shared with the other phages of *Septuagintavirus* genus. The exceptions are W70_129, encoding a structural protein of unknown function not shared with phages A5-4 and A7-1, and W70_239 encoding a structural protein

with the predicted role as a chromosome condensation protein not shared with phage A7-1. Other genera of *Vequintavirinae* subfamily have also had their virion proteins identified by mass spectrometry; these include *Escherichia* phage rV5 of *Vequintavirus* (32), *Salmonella* phage PVP-SE1 of *Seunavirus* (20), and *Pectobacterium* phage CB7 of *Certrevirus* (24). Comparisons of the structural proteome of these phages with that of phage W70 showed that they share 19 structural proteins involved in capsid, tail, and baseplate morphogenesis (Table S6). However, the greatest homology was found between phages W70 and *Pectobacterium* phage CB7 (22 proteins), sharing two OGs (W70_107 and W70_112) implicated in the formation of tail fibers not associated with *Escherichia* phage rV5 and *Salmonella* phage PVP-SE1. The formation of tail fibers can play a crucial role in the mechanism of infection of phages. The observed similarities in proteins involved in tail fiber formation between the *Septuagintavirus* and *Certrevirus* genera suggest that the infection strategy, particularly in their ability to bind to the host's specific receptors, could be similar. Notably, the narrow host range characteristic of *Septuagintavirus* genera, as described in the section "Isolation of *Escherichia* Phages and Their Characterization", is also reported in characterized phages belonging to the *Certrevirus* genus, such as *Pectobacterium* phage CB7 (24), phage PBES02 (33), or phage P7_Pc (34). Previous studies have indicated that *Certrevirus* phages bind to flagella located on the surface of the host cell (24), suggesting a potential shared mechanism with *Septuagintavirus*. Another phage from the *Vequintavirinae* subfamily with a narrow lytic host range is phage Hena1 (*Henunavirus* genus), which encodes a putative exopolysaccharide (EPS) depolymerase, that may be involved in the degradation of EPS, facilitating phage attachment to the bacterial cell wall (23). However, most genera within the subfamily *Vequintavirinae* show a broad host range, interacting primarily with the host cell via the outer membrane, specifically through lipopolysaccharide (LPS) and/or other surface receptors. In addition, these phages potentially can attach to multiple surface receptors facilitating their broad host range such as has been described for phage Av-05 (*Avunavirus* genus) that infects different *E. coli* and *Salmonella* strains (21), PVP-SE1 phage (*Seunavirus* genus) with a broad lytic spectrum against *Salmonella* strains (20), or phage Rv5 (*Vequintavirus* genus) with a broad *E. coli* strains spectrum (32). Further analysis would be necessary to confirm the specific mechanisms underlying the infectivity of these newly described *E. coli* phages.

## DNA replication, nucleotide metabolism and methylation

The four phages isolated in this study shared several genes encoding products expected to be involved in DNA replication (Table S5), such as three OGs likely to act as DNA polymerases and four OGs possessing domains relating to helicase or primase activity. However, diversity was observed among phages, as they do not universally share these proteins. For instance, an additional DNA polymerase was identified only in A5-4, and different DNA helicases were exclusively detected in A73 (one) and W70 (two). This observation suggests variations in the strategies these phages employ for replicating their genomes. Phages universally share proteins, such as DNA ligase and recombination endonuclease VII of *Escherichia* phage T4. The latter protein is involved in the repair of DNA mismatch and packaging by removing branched replicative DNA (35).

The genomes of the here described phages encode proteins related to nucleotide metabolism (Table S5). They possess aerobic class I ribonucleotide reductase (RNR) subunits that catalyze the reduction of nucleoside triphosphate to deoxynucleoside triphosphates (dNTPs) in aerobic conditions. Moreover, these phage genomes featured the anaerobic class III ribonucleotide reductase subunits, facilitating dNTP production under anaerobic conditions. Additionally, their genomes encode glutaredoxin to support the RNR class I function by aiding in RNR reduction (36). They also possess the ability to impact the deoxythymidine monophosphate (dTMP) pool of the host by a deoxyuridine monophosphate (dUMP) through their thymidylate synthase (IPR036098).

Two types of DNA methylase were discernible within these phages (Table S5), suggesting that their genomes probably harbor methylated adenine and cytosine. This

inference is based on the presence of a presumed N-6 adenine-specific DNA methylase (IPR002052) and a DNA cytosine methylase identified through HHpred analysis (best hit: Cytosine-specific methyltransferase, PDB accession no. 3ME5_A).

## Cell wall-degrading enzymes and cell lysis proteins

Peptidoglycan-degrading enzymes play a crucial role in the early stages of phage infection by facilitating the penetration of the host cell wall during the injection of phage DNA, a process carried out by virion-associated lysins (37). Furthermore, these enzymes are involved in host cell lysis at the conclusion of the phage lytic cycle. These phages are equipped with an endopeptidase belonging to the peptidase M15C domain (IPR039561), specifically designed to target the peptide cross-links within peptidoglycan (Table S5). In addition to this, they also possess a putative cell wall hydrolase, predicted to be an N-acetylmuramyl-L-alanine amidase, resembling SleB. SleB is a protein found in *Bacillus subtilis* that is responsible for hydrolyzing the spore cortex during germination (IPR011105) (38). Notably, this gene product was also identified in phages of the *Certrevirus* genus (24).

## Conclusion

Here, we describe the isolation and characterization of *Escherichia* phages W70, A5-4, A7-1, and A73, representing a novel genus of the subfamily *Vequintavirinae*, which we named *Septuagintavirus*. The genus name was derived from the name given to the first isolated example of these phages, namely *Escherichia* phage W70. These phages are close relatives of those of *Certrevirus,* sharing several features found among members of this genus. This includes their genome-encoding proteins with similar functions in relation to DNA replication, nucleotide metabolism, DNA methylation, and host lysis, as previously reported (24). Additionally, the analysis of the structural proteins of *Escherichia* phage W70 indicates that their virions are expected to be more similar to those of *Certrevirus* than other genera of *Vequintavirinae*, namely *Vequintavirus* and *Seunavirus*, especially concerning proteins implicated in the formation of tail fibers. In this study, the host range determined for *Septuagintavirus* phages was found to be narrow, with phages collectively infecting only six of the sixteen tested strains of *E. coli*. Phage tail fibers mediate phage binding to a specific receptor on the cognate bacterial host surface. The resemblance between the tail fibers of *Certrevirus* phages and those of *Septuagintavirus* suggests a potentially shared utilization of similar receptors. Notably, *Certrevirus* phages use flagella as their host cell receptors and have been described to possess a narrow host range (24, 33, 34). This observation may indicate a parallel mechanism in the host recognition process between these two genera. However, further experimental work will be necessary to confirm this conclusion.

## MATERIALS AND METHODS

### Bacterial strains and culturing conditions

Seventeen *E. coli* strains were used in this study, as outlined in Table S1. Frozen stocks of bacterial strains were stored in 30% glycerol at −80°C with these routinely streaked directly onto Lysogeny Broth (LB) (Merck, Darmstadt, Germany) agar plates (1.5% agar wt/vol), followed by aerobic incubation at 37°C for 24 hours. Liquid cultures were prepared by inoculating single colonies into 10 mL of LB medium and incubating them aerobically at 37°C with agitation for 24 hours. The phylotype of the *E. coli* strains used in this study was determined using a phylotyping PCR as previously described (1).

### Isolation, purification, and propagation of the novel *E. coli* phages

Phage isolation utilized an enrichment procedure using wastewater or animal feces collected from a single multi-species farm in Munster (Ireland). One gram of the sample

was aliquoted and suspended in 5 mL of SM buffer (50 mM Tris-HCl pH 7.5, 8 mM MgSO$_4$, 100 mM NaCl). The wastewater samples and fecal solutions were centrifuged at 4,500 × $g$ for 15 minutes at 4°C (Megafuge 16 centrifuge, ThermoFisher Scientific, Waltham, MA, USA), and the supernatant was filtered through a 0.45 µm pore syringe-mounted polyethersulfone (PES) membrane filter (Sarstedt, Nümbrecht, Germany). Then, 2 mL of the filtrate was placed into 10 mL of LB broth with 0.5 mL of overnight culture of *E. coli* K12 MG1655 (the host) and was incubated aerobically overnight at 37°C with shaking. After incubation, the samples were centrifuged at 4,500 × $g$ for 15 minutes at 4°C to pellet the bacteria, and the supernatant was filtered through 0.45 µm PES membrane filters. Then 2 mL of the filtrate was mixed with 2 mL of LB soft agar (0.4% agar wt/vol) and 0.5 mL of an overnight culture of *E. coli K12* MG1655 to create an overlay, which was spread onto LB agar plates (1.5% agar wt/vol). These were then aerobically incubated for 24 hours at 37°C to assess plaque formation. Individual and different plaques were cut out and soaked in 100 µL of SM buffer for two hours with intermittent gentle vortexing to facilitate the release of phages from the soft agar. The phage solution was filtered through a 0.45 µm pore PES membrane filter and re-plated in the same manner as previously mentioned. This process was repeated twice more to generate pure phage stocks (39). Phages were propagated using *E. coli* K12 MG1655 through the overlay method, as previously described (39). Specific primers targeting the potential new phages identified after genome sequencing were designed using the Primer-BLAST tool (https://www.ncbi.nlm.nih.gov/tools/primer-blast/) (Table S2) to verify the distinctiveness of all detected phages. The primers were synthesized by Sigma-Aldrich (Sigma-Aldrich, MO, USA).

## Host range analysis

Sixteen different *E. coli* strains obtained from the feces of individuals with IBD, isolated and utilized in a previous study (40), were employed to carry out a host range analysis of the isolated phages. These strains were selected based on their phylotype classification to encompass a broad spectrum of distinct phylogenetic clades (A, B1, B2, D, and E) (Table S1). Additionally, other genera from the family Enterobacteriaceae—Klebsiella *pneumoniae*, *Enterobacter aerogenes*, *Salmonella enterica* serovar typhimurium, *Proteus vulgaris,* and *Citrobacter braakii*—were also tested. The host range of each phage was assessed by spotting serial dilutions in SM buffer (10 µL) of the phage, ranging from undiluted to 10$^{-8}$, onto the top of an LB overlay previously seeded with 500 µL of each overnight culture on LB plates (1.5% agar wt/vol). The plates were incubated overnight at 37°C. Bacterial strains were classified as either resistant or sensitive based on the observation of plaque formation. An *in silico* prediction of host range was also conducted.

## Bacteriophage kill-curves

Phage lytic activity and formation of phage-resistant bacteria were evaluated through bacteriophage kill curves, following the method described elsewhere (40). Briefly, fresh LB medium was inoculated with 5% of an overnight culture of the host strain and allowed to grow to ~1 × 10$^8$ cfu/mL. Serial dilutions of phages in SM buffer were individually added to the culture to obtain a range of multiplicity of infection (MOI) values from 0.001 to 10. The mixture was placed in the wells of a flat-bottom 96-well micro test plate (Sarstedt, Nümbrecht, Germany), sealed, and incubated at 37°C with shaking. The kill curves were determined by measuring the OD$_{595}$ every 15 minutes for 9.5 hours using a microtiter plate reader (Multiskan FC, ThermoScientific, Waltham, MA, USA). All plates included a negative (SM buffer and LB medium) and a positive control (bacterial culture). All samples were analyzed in triplicate.

## Transmission electron microscopy

Transmission electron microscopy (TEM) was used to visualize the isolated *E. coli* W70 phage. Before the analysis, the phage was purified using a CsCl (Sigma Aldrich, St Louis, MO, USA) density gradient as follows. 180 mL of a high-titer W70 phage filtrate (~ $10^{10}$–$10^{11}$ pfu/mL) was concentrated through ultracentrifugation (F65L-6 × 13.5 rotor, ThermoFisher Scientific, Waltham, MA, USA) in three rounds of 2 hours centrifugations at 105,000 × *g* at 4°C. The obtained pellets were resuspended in SM buffer, collecting a final volume of 6 mL. 3 mL of the concentrated phage were layered onto a step gradient containing CsCl with densities of 1.7 g/cm$^3$, 1.5 g/cm$^3$, and 1.3 g/cm$^3$ (from bottom to top), followed by ultracentrifugation at 105,000 × *g* at 4°C for three hours. The phage band was extracted with a sterile syringe and dialyzed against 1 L of SM buffer using Amicon Ultra-0.5 Centrifugal Filter Units with a 3 kDa MWCO membrane (Merck Millipore, Burlington, MA, USA) overnight at 4°C in fresh SM buffer. The purified phage was adsorbed to freshly prepared ultra-thin carbon film for 20 minutes. Subsequently, it was negatively stained with 2% (wt/vol) uranyl acetate and analyzed using a Talos L120C transmission electron microscope (ThermoFisher, Eindhoven, The Netherlands), operating at an acceleration voltage of 80 kV. Digital micrographs were acquired with a 4k × 4k Ceta camera (ThermoFisher, Eindhoven, The Netherlands) and enhanced (brightness and contrast) in GIMP (v2.10). ImageJ software was used to measure phage dimensions, and the average values were obtained from a minimum of 10 images (41).

## Phage DNA extraction and genome sequencing

Filtered phage lysates were treated with 15% (wt/vol) polyethylene glycol (PEG-8000) (Sigma Aldrich, St Louis, MO, USA) along with 1M NaCl, and incubated overnight at 4°C. Subsequently, the samples were centrifuged at 11,000 × *g* for 60 minutes at 4°C. The pellets were resuspended in SM buffer, and treatments with DNase I (ThermoFisher Scientific, Vilnius, Lithuania) and RNase I (ThermoFisher Scientific, Vilnius, Lithuania) were performed. Phage genome DNA extraction was carried out using the phage DNA isolation kit (Norgen Biotek, Ontario, Canada), followed by a cleanup procedure using the Wizard DNA cleanup system (Promega, Madison, WI, USA), following the manufacturer's indications. DNA quantification was performed using the Qubit dsDNA Broad-Range assay kit (ThermoFisher Scientific, Vilnius, Lithuania) before standardizing paired-end Nextera XT library preparation (Illumina, San Diego, CA, USA). Library quality was examined with the Agilent Technology 2100 Bioanalyzer (Agilent Technologies, Waldbronn, Germany) with a High Sensitivity DNA chip, following the manufacturer's recommendations, before sequencing on an Illumina MiSeq platform (Illumina, San Diego, CA, USA).

## Genomic and phylogenetic analysis

Genome *de novo* assembly of phages was carried out using default parameters with metaSPAdes (42). Open reading frames (ORFs) were predicted using the RAST server (43). Further analyses of the ORFs were conducted with BLASTp against the PHASTER database (44, 45), as well as Interproscan (46) and HHpred (47). Translated ORFs from the phages were searched against hidden Markov model profiles obtained from the Prokaryotic Virus Orthologous Groups (pVOGs) (48) and Prokaryotic Virus Remote Homologous Groups (PHROG) (49) databases using HMMER3 hmmscan (v3.4) (50) with an E-value cut-off of $1 \times 10^{-5}$. The presence of transfer RNA genes was investigated using ARAGORN (v1.2.36) (51). Genomic comparisons between phages were conducted with tBLASTx and visualized using Easyfig (v2.2.5) (31). Total proteome comparisons between phages were conducted with CoreGenes3.5 using a BLASTP threshold of 75% (52). Phage genome nucleotide similarity was calculated using VIRIDIC (29). VICTOR was employed for all pairwise comparisons between phages at the amino acid level using the Genome-BLAST Distance Phylogeny (GBDP) method (30). With branch support, the phylogenetic trees were rooted at the midpoint and visualized with iTOL (v4) (53). The Proksee tool was used to generate the genomic map of the phage W70 genome (54). *In*

*silico* prediction of host range of phages of this study was examined using iPHoP (v1.3.3) (55).

## Proteome analysis of virion structural proteins

A purified high-titer stock of the W70 phage (~ $10^{10}$–$10^{11}$ pfu/mL) was prepared to analyze virion structural proteins, following the methodology outlined in the TEM analysis section (Transmission Electron Microscopy) with some modifications. In this case, the CsCl gradient was replaced by a sucrose step gradient with densities of 45%, 35%, 20%, and 10% (from bottom to top) in TM buffer (50 mM Tris-HCl pH 7.5, 10 mM $MgCl_2$). 3 mL of the phage solution was layered onto the top of the gradient, and was ultracentrifuged at 70,000 × $g$ for 40 minutes at 4°C. The resulting phage band was extracted using a blunt cannula, resuspended in 10 mL of ice-cold TM buffer, and pelleted by ultracentrifugation at 145,000 × $g$ for 2 hours at 4°C. The pellet was resuspended in 500 µL of TM buffer and incubated at 4°C overnight. The proteome sample preparation and analysis were carried out as described elsewhere (56), with minor modifications. The isolated phages were subjected to lysis buffer [50 mM Tris-HCl pH 7.5, 1% sodium lauryl sulfate (SLS), 2 mM Tris(2-carboxyethyl)phosphine (TCEP)], heated to 95°C for 10 minutes, and sonicated (10 seconds, 20% amplitude, 0.5 pulse) to degrade nucleic acids. Following this, iodoacetamide (4 mM) was added, and the samples were incubated for 30 minutes under light protection. The proteins were precipitated using acetone. The resulting pellets were washed with 500 µL of methanol (−80°C), air-dried, and resuspended in 50 µL of resuspension buffer (50 mM Tris-HCl pH 7.5, 0.5% SLS). The protein concentration was determined using the BCA assay (Pierce TM, BCA protein assay kit, ThermoFisher Scientific, Waltham, MA, USA). To 10 µg of isolated proteins, 0.5 µg of sequencing-grade trypsin (Promega) was added. Digestion was carried out overnight at 30°C. The remaining SLS was precipitated by adding 1.5% trifluoroacetic acid (TFA) (vol/vol) and centrifugated at 4°C at 17,000 × $g$ for 10 minutes. The supernatant was desalted using C18 solid-phase columns (Chromabond C18 spin columns, Macherey Nagel, Düren, Germany), and the solvent was removed by evaporation. Dried peptides were stored at −20°C until further use.

Dried peptides were reconstituted in 0.1% TFA and analyzed using liquid-chromatography-mass spectrometry (LC-MS) on an Exploris 480 instrument connected to an Ultimate 3000 RSLC nano and a nanospray flex ion source (all Thermo Scientific). Peptide separation was performed on a reverse phase HPLC column (75 µm × 42 cm) packed in-house with C18 resin (2.4 µm; Dr. Maisch) using the following separating gradient: 98% solvent A (0.15% formic acid) and 2% solvent B (99.85% acetonitrile, 0.15% formic acid) to 30% solvent B over 45 minutes at a flow rate of 300 nL/min. The data acquisition mode was set to obtain one high-resolution MS scan at a resolution of 60,000 full width at half maximum (at *m/z* 200) followed by MS/MS scans of the most intense ions within 1 s (cycle 1 s). The charged state screening modus was enabled to exclude unassigned and singly charged ions to increase the efficiency of MS/MS attempts. The dynamic exclusion duration was set to 14 seconds. The ion accumulation time was set to 50 ms (MS) and 50 ms at 17,500 resolution (MS/MS). The automatic gain control (AGC) was set to 3 × $10^6$ for MS survey scan and 2 × $10^5$ for MS/MS scans. For spectral-based assessment, MS raw files searches were carried out using MSFragger embedded within Scaffold 4 (Proteome Software) with 20 ppm peptide and fragment tolerance with carbamidomethylation (C) as fixed, and oxidation (M) as variable modification using a uniprot protein database. A cut-off of <25 peptides for total spectrum count was used to exclude low-level contaminating proteins not associated with phage virion.

Sodium dodecyl sulfate polyacrylamide gel electrophoresis (SDS-PAGE) was performed to characterize the separation of W70 phage proteins in a gel based on their molecular weight. To achieve this, a high-titer stock of the W70 phage, obtained using a CsCl gradient (section "Transmission Electron Microscopy"), was subjected to a chloroform-methanol treatment to induce protein precipitation. Four volumes of methanol (Sigma Aldrich, St Louis, MO, USA) were added to one volume of sample.

The mixture was vortexed and centrifuged for 20 seconds at 5,300 × $g$. Two volumes of chloroform (Fisher Scientific, Loughborough, England, UK) were added and centrifuged for 20 seconds at 5,300 × $g$. Then, three volumes of distilled water were added to facilitate phase separation. The mixture was gently vortexed and was centrifuged at 10,000 × $g$ for 15 minutes. The top aqueous layer was discarded with the same volume of methanol added to the remaining sample and gently vortexed, and the mixture was centrifuged for 10 minutes at 5,300 × $g$, resulting in an opaque protein pellet. This protein pellet was subjected to SDS-PAGE gel electrophoresis as described previously (57).

## ACKNOWLEDGMENTS

We would like to thank Rob Lavigne at the KU Leaven, Belgium, for his advice that enabled the preparation of this manuscript.

This publication had the financial support of Science Foundation Ireland (SFI) under grant numbers SFI/12/RC/2273_P2, SFI/15/ERCD/3189, and SFI/14/SPAPC/B3032, and a research grant from Janssen Biotech. K. H. is supported by funding from the Max Planck Society and the German Research Council (DFG SPP 2330, Project number 464500427).

A.C.-M. performed wet lab work, analyzed results, and wrote the original manuscript; C.B. conducted bioinformatics analysis, analyzed results, and wrote the original manuscript; N.P. and K.H. performed proteomics analysis; F.H. and C.M.A.P.F conducted transmission electronic microscopy analysis; H.S and A.S.B. performed wet lab work; L.A.D. managed the project; A.N.S., R.P.R., and C.H. secured funding and supervised the project. All authors reviewed and edited the manuscript.

## AUTHOR AFFILIATIONS

[1]APC Microbiome Ireland and School of Microbiology, University College Cork, Cork, Ireland

[2]Max-Planck-Institute for Terrestrial Microbiology, Marburg, Germany

[3]Department of Microbiology and Biotechnology, Max Rubner-Institut, Kiel, Germany

[4]Center for Synthetic Microbiology (SYNMIKRO), Marburg, Germany

[5]School of Microbiology, University College Cork, Cork, Ireland

## AUTHOR ORCIDs

Adrián Cortés-Martín http://orcid.org/0000-0001-5542-4984

Colin Buttimer http://orcid.org/0000-0002-3361-8902

R. Paul Ross http://orcid.org/0000-0003-4876-8839

Colin Hill http://orcid.org/0000-0002-8527-1445

## FUNDING

| Funder | Grant(s) | Author(s) |
| --- | --- | --- |
| Science Foundation Ireland (SFI) | SFI/12/RC/2273_P2,SFI/15/ERCD/3189,SFI/14/SPAPC/B3032 | R. Paul Ross<br>Colin Hill |
| Janssen Biotech Inc. | SFI/14/SPAPC/B3032 | R. Paul Ross<br>Colin Hill |
| Max Planck Society and the German Research Caouncil | DFG SPP 2330 | Katharina Höfer |

## DATA AVAILABILITY

The genomes of phages described in this study are available at GenBank under the following accession numbers: OP795442 (A7-1), OP744025 (A5-4), OP778609 (A73), and OP778610 (W70).

## ADDITIONAL FILES

The following material is available online.

### Supplemental Material

**Supplemental figures (Spectrum00592-24-s0001.docx).** Fig. S1 to S3.
**Supplemental tables (Spectrum00592-24-s0002.xlsx).** Tables S1 to S6.

### Open Peer Review

**PEER REVIEW HISTORY (review-history.pdf).** An accounting of the reviewer comments and feedback.

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
