## [Reviewer comments · Microbiology Spectrum]

Microbiology Spectrum

Isolation and characterization of *Septuagintavirus*; a novel clade of *Escherichia coli* phages within the subfamily *Vequintavirinae*

Adrián Cortés-Martín, Colin Buttimer, Nadiia Pozhydaieva, Frank Hille, Hiba Shareefdeen, Andrei Bolocan, Lorraine Draper, Andrey Shkoporov, Charles Franz, Katharina Höfer, R. Paul Ross, and Colin Hill

Corresponding Author(s): Colin Hill, University College Cork

Review Timeline:

Submission Date:	March 6, 2024
Editorial Decision:	April 29, 2024
Revision Received:	June 5, 2024
Accepted:	June 25, 2024

Editor: Thomas Denes

Reviewer(s): Disclosure of reviewer identity is with reference to reviewer comments included in decision letter(s). The following individuals involved in review of your submission have agreed to reveal their identity: Sanghamitra Saha (Reviewer #1)

Transaction Report:

DOI: <https://doi.org/10.1128/spectrum.00592-24>

Re: Spectrum00592-24 (Isolation and characterization of *Septuagintavirus*; a novel clade of *Escherichia coli* phages within the subfamily *Vequintavirinae*)

Dear Prof. Colin Hill:

Thank you for the opportunity to review your work. Below you will find my comments, instructions from the Spectrum editorial office, and the reviewer comments.

Overall, this manuscript was favorably reviewed by two reviewers; although they did leave some comments that I would be interested in seeing your response to. I look forward to seeing your response and any revisions.

Revision Guidelines

Sincerely,
Thomas Denes
Editor
Microbiology Spectrum

Reviewer #1 (Comments for the Author):

The authors describe a new clade of Ecoli phages and named it Septuagintavirus within the subfamily, Vequintavirinae. They isolated four novel Ecoli phages (W70, A7-1, A5-4, and A73) that infect E.coli K12 MG1655. This study and discovery of novel

phages offers an alternative treatment for GI and other diseases in which E. coli maybe involved. With the emergence of antibiotic resistance strains. E.coli phages offer the option of using phages as therapeutic agents. The isolation of lytic phages, which the above-mentioned phages are, are an important step in that direction. The phages were isolated from wastewater and animal feces and produced clear plaques. The phages also showed some lytic activity against other E. coli strains. The authors did a lysis profile of all four phages and identified them to be myovirus based on electron microscopy. The genomes of phages ranged from 137,323 and 145, 365 bp and showed a 90% homology amongst them. Genome annotation assigned functions to about 23% of the genes. Using phylogenetic analysis VICTOR, it was deduced that these phages form a distinct clade within Vequentevirinae and are placed close to Certrevirus genus.

The different classes of genes and their assigned functions were discussed; these include structural proteins (proteome analysis was done by SDS PAGE and mass spectrometry), DNA replication, cell wall degrading enzymes and cell lysis.

The authors conclude by stating that these phages are close relatives of Certevirus. The resemblance of tail fibers between Septuagintavirus and Certevirus suggest a shared utilization of cell surface receptors.

Was curious to know if these phages would be effective against other bacterial stains that are part of the human microbiome.

Reviewer #2 (Comments for the Author):

The author isolated four E.coli phages, analyzed their host ranges, structural proteins, and morphologies, and concluded that these phages belong to a novel genus, Septuagintavirus, based on genome analysis.

Main document:

1. Figure S1 only displays plaques produced by phage W70. Why are there no images of plaques from the other three phages?

The document mentions only spot assays for these phages without showcasing their individual plaque morphologies.

2. Lines 122-123 state, "lower MOIs appeared to result in less efficient recovery than higher MOIs." What could be the underlying reason for this observation?

3. Are there electron microscope images available for the other three phages?

4. What are the pH and temperature sensitivities of the other three phages?

5. Line 134 mentions, "with a shared nucleotide similarity between genomes exceeding 90%." What is the exact nucleotide similarity? Additionally, which phage in the NCBI database shares the highest homology with these four phages?

6. Lines 155-156: Among the Certrevirus phages, which one shares the highest homology with W70? What is the nucleotide similarity?

7. The author concludes that the four phages belong to a distinct subfamily, Vequentavirinae, solely based on VICTOR analysis. Is there any other evidence to support this classification?

Comments and Suggestions for the authors:

The authors describe a new clade of *E. coli* phages and named it Septuagintavirus within the subfamily, Vequentavirinae. They isolated four novel *E. coli* phages (W70, A7-1, A5-4, and A73) that infect *E. coli* K12 MG1655. This study and discovery of novel phages offers an alternative treatment for GI and other diseases in which *E. coli* may be involved. With the emergence of antibiotic resistance strains, *E. coli* phages offer the option of using phages as therapeutic agents. The isolation of lytic phages, which the above-mentioned phages are, are an important step in that direction. The phages were isolated from wastewater and animal feces and produced clear plaques. The phages also showed some lytic activity against other *E. coli* strains. The authors did a lysis profile of all four phages and identified them to be myovirus based on electron microscopy. The genomes of phages ranged from 137,323 and 145,365 bp and showed a 90% homology amongst them. Genome annotation assigned functions to about 23% of the genes. Using phylogenetic analysis VICTOR, it was deduced that these phages form a distinct clade within Vequentavirinae and are placed close to Certevirus genus.

The different classes of genes and their assigned functions were discussed; these include structural proteins (proteome analysis was done by SDS PAGE and mass spectrometry), DNA replication, cell wall degrading enzymes and cell lysis.

The authors conclude by stating that these phages are close relatives of Certevirus. The resemblance of tail fibers between Septuagintavirus and Certevirus suggest a shared utilization of cell surface receptors.

Was curious to know if these phages would be able to infect against other bacterial stains that are part of the human microbiome.

Confidential remarks for the editor

The paper discusses the discovery of four novel *E. coli* phages and the discussion of the potential functions of the genes. The experimental findings are presented well and are comprehensive. I did not find any grammatical and/or language issues throughout the paper; I think it was extremely well written.

According to me, this paper is better suited for Microbiology Resource Announcements where we have published some papers. The discovery and annotation of phages is routinely featured in that journal.

Reviewer #1 (Comments for the Author): The authors describe a new clade of E. coli phages and named it Septuagintavirus within the subfamily, Vequentavirinae. They isolated four novel E. coli phages (W70, A7-1, A5-4, and A73) that infect E. coli K12 MG1655. This study and discovery of novel phages offers an alternative treatment for GI and other diseases in which E. coli maybe involved. With the emergence of antibiotic resistance strains. E. coli phages offer the option of using phages as therapeutic agents. The isolation of lytic phages, which the above-mentioned phages are, are an important step in that direction. The phages were isolated from wastewater and animal feces and produced clear plaques. The phages also showed some lytic activity against other E. coli strains. The authors did a lysis profile of all four phages and identified them to be myovirus based on electron microscopy. The genomes of phages ranged from 137,323 and 145,365 bp and showed a 90% homology amongst them. Genome annotation assigned functions to about 23% of the genes. Using phylogenetic analysis VICTOR, it was deduced that these phages form a distinct clade within Vequentavirinae and are placed close to Certrevirus genus.

The different classes of genes and their assigned functions were discussed; these include structural proteins (proteome analysis was done by SDS PAGE and mass spectrometry), DNA replication, cell wall degrading enzymes and cell lysis.

The authors conclude by stating that these phages are close relatives of Certrevirus. The resemblance of tail fibers between Septuagintavirus and Certrevirus suggest a shared utilization of cell surface receptors.

Was curious to know if these phages would be effective against other bacterial stains that are part of the human microbiome.

We appreciate the reviewer's comments on our work. Our results indicate that these phages have a narrow host range, as they only produced plaques in a few *E. coli* strains from an *E. coli* panel, suggesting they are highly strain/species-specific. Following the reviewer's suggestion, we have tested other gut bacterial species from the family Enterobacteriaceae to assess the phages' effectiveness against them (Table S3). However, we have not observed plaque formation in any of these species. Nonetheless, we cannot guarantee whether these phages would be effective against other bacterial strains from the human microbiome that have not been tested. To delve into this question, we have performed an *in-silico* analysis using the iPHoP tool, which predicts the host genus of phages based on their genome sequences. This tool infers candidate phage-host pairs integrating results from multiple host prediction approaches. Results have been added in a new Supplementary Table (Table S4).

Reviewer #2 (Comments for the Author): The author isolated four E. coli phages, analyzed their host ranges, structural proteins, and morphologies, and concluded that these phages belong to a novel genus, Septuagintavirus, based on genome analysis.

Main document:

1. Figure S1 only displays plaques produced by phage W70. Why are there no images of plaques from the other three phages? The document mentions only spot assays for these phages without showcasing their individual plaque morphologies.

In agreement with the reviewer's suggestion, we have included images of the plaques for each of the phages in the Figure S1.

2. Lines 122-123 state, "lower MOIs appeared to result in less efficient recovery than higher MOIs." What could be the underlying reason for this observation?

The potential reason for this observation could be that kill curve assays with lower MOIs experienced more aggressive bacterial growth at the initial stages of the experiment (1-2 hours after starting), prior to lysis and bacterial recovery. The limited growth observed is likely due to the prior growth of bacteria having exhausted nutrients available in media, thus limiting bacterial recovery post-lysis.

3. Are there electron microscope images available for the other three phages?

Unfortunately, there are no electron microscope images available for the other three phages. Considering the high nucleotide similarity among the four phages, all belonging to the same genus, we did not anticipate big differences in morphology between them. Therefore, we deemed it sufficient to examine only a representative phage of this collection.

4. What are the pH and temperature sensitivities of the other three phages?

The pH and temperature sensitivities for the phages have not been explored in this study as these phages are unlikely to be suitable for phage therapy applications due to their extremely narrow host range. We acknowledge the potential importance of investigating these factors for broader scientific understanding, but for the purpose of this manuscript, which focuses on describing this new genus, the specific relevance of this knowledge remains uncertain.

5. Line 134 mentions, "with a shared nucleotide similarity between genomes exceeding 90%." What is the exact nucleotide similarity? Additionally, which phage in the NCBI database shares the highest homology with these four phages?

Please refer to Figure 3 for the nucleotide similarity values of these phages. The Figure shows that these values range between 90-95% between the four phages. Also, the phages exhibiting the highest homology with these four phages are those of *Certrevirus* genus, with nucleotide similarity values ranging between 28-30%; please see Figure 3 for further details.

6. Lines 155-156: Among the *Certrevirus* phages, which one shares the highest homology with W70? What is the nucleotide similarity?

This information is also provided in Figure 3, showing the nucleotide similarity between the different members of the *Vequintavirinae* family and the newly isolated phages. The *Certrevirus* phage with the highest homology to W70 was *Pectobacterium* phage phiTE, with a 30% of similarity. Please, refer to the Figure 3 for further information.

7. The author concludes that the four phages belong to a distinct subfamily, *Vequintavirinae*, solely based on VICTOR analysis. Is there any other evidence to support this classification?

In this work we are describing the creation of a new genus, *Septuagintavirus*, within the subfamily of *Vequintavirinae*. The position of this clade is based on genome nucleotide similarity of phages and their positions in phylograms based on total proteome homology as determined by the tool VICTOR. Please note that this proposed genus has been submitted to the ICTV and recently approved. Please refer to the following link for the documentation related to this genus proposal:

https://ictv.global/taxonomy/taxondetails?taxnode_id=202319268&taxon_name=Septuagintavirus%20A73

Re: Spectrum00592-24R1 (Isolation and characterization of *Septuagintavirus*; a novel clade of *Escherichia coli* phages within the subfamily *Vequintavirinae*)

Dear Prof. Colin Hill:

Your manuscript has been accepted, and I am forwarding it to the ASM production staff for publication. Your paper will first be checked to make sure all elements meet the technical requirements. ASM staff will contact you if anything needs to be revised before copyediting and production can begin. Otherwise, you will be notified when your proofs are ready to be viewed.

Sincerely,
Thomas Denes
Editor
Microbiology Spectrum